# The socioeconomic impact of orthopaedic trauma: A systematic review and meta-analysis

**Nathan N. O'Hara**[1,2]*, **Marckenley Isaac**[1], **Gerard P. Slobogean**[1], **Niek S. Klazinga**[2]

**1** Department of Orthopaedics, University of Maryland School of Medicine, Baltimore, Maryland, United States of America, **2** Department of Public Health, Amsterdam Public Health Research Institute, Amsterdam UMC, University of Amsterdam, Amsterdam, the Netherlands

* nohara@som.umaryland.edu

**Data Availability Statement:** All relevant data are within the manuscript and its Supporting Information files.

**Funding:** The authors received no specific funding for this work.

## Abstract

The overall objective of this study was to determine the patient-level socioeconomic impact resulting from orthopaedic trauma in the available literature. The MEDLINE, Embase, and Scopus databases were searched in December 2019. Studies were eligible for inclusion if more than 75% of the study population sustained an appendicular fracture due to an acute trauma, the mean age was 18 through 65 years, and the study included a socioeconomic outcome, defined as a measure of income, employment status, or educational status. Two independent reviewers performed data extraction and quality assessment. Pooled estimates of the socioeconomic outcome measures were calculated using random-effects models with inverse variance weighting. Two-hundred-five studies met the eligibility criteria. These studies utilized five different socioeconomic outcomes, including *return to work* (n = 119), *absenteeism days from work* (n = 104), *productivity loss* (n = 11), *income loss* (n = 11), and *new unemployment* (n = 10). Pooled estimates for return to work remained relatively consistent across the 6-, 12-, and 24-month timepoint estimates of 58.7%, 67.7%, and 60.9%, respectively. The pooled estimate for mean days absent from work was 102.3 days (95% CI: 94.8–109.8). Thirteen-percent had lost employment at one-year post-injury (95% CI: 4.8–30.7). Tremendous heterogeneity ($I^2$>89%) was observed for all pooled socioeconomic outcomes. These results suggest that orthopaedic injury can have a substantial impact on the patient's socioeconomic well-being, which may negatively affect a person's psychological wellbeing and happiness. However, socioeconomic recovery following injury can be very nuanced, and using only a single socioeconomic outcome yields inherent bias. Informative and accurate socioeconomic outcome assessment requires a multifaceted approach and further standardization.

## Introduction

Orthopaedic trauma is a common reason for ongoing pain and significant disability [1,2]. The resumption of work activities following injury has been demonstrated to be a reliable marker

**Competing interests:** The authors have declared that no competing interests exist.

of healing and is significantly associated with increased patient satisfaction [3,4]. For these reasons, outcomes, such as return to work and absenteeism days from work, are important dimensions in determining value-based healthcare [5].

Socioeconomic outcomes can be broadly defined as events related to income, employment, and education [6]. It has been suggested that efforts to mitigate income loss have the potential to reduce the severity and costs of major diseases more than traditional medical advances [7]. Socioeconomic measures are particularly relevant for extremity fracture patients, as the injuries commonly afflict the working age population and the injuries themselves are frequently work-related [8]. A better understanding of the socioeconomic consequences of fractures will aid in advocating for the necessary resources and reimbursements to appropriately manage these injuries and mitigate negative socioeconomic outcomes.

The overall objective of this study was to determine the socioeconomic impact of orthopaedic trauma in the available literature. We aimed to achieve this objective by defining the various socioeconomic outcome measures and calculating pooled socioeconomic outcomes for extremity fracture patients at commonly reported time points. Finally, the study aimed to identify common limitations in the use of socioeconomic outcome measures for extremity fracture research.

## Materials and methods

The systematic review protocol was developed based on the Preferred Reporting Items for Systematic Review and Meta-analysis guidelines (PRISMA) and registered in PROSPERO (CRD42018093622) [9].

### Eligibility criteria

Studies were eligible for inclusion if more than 75% of the study population sustained an appendicular fracture due to an acute trauma, the mean age of the study population was between 18 and 65 years of age, and the study included a socioeconomic outcome, defined as a measure of income, employment status, or educational status. Studies were excluded if over half of the study population was greater than 65 years of age, had pathologic fractures (osteoporotic, osteomyelitis), had a spinal injury or traumatic brain injury, or a traumatic amputation. In addition, we excluded case series of less than ten study participants, as well as expert opinion and narrative papers.

### Identification of studies

An experienced academic research librarian conducted searches in MEDLINE (Ovid), Embase (Elsevier), and Scopus on December 3, 2019, without restrictions on publication date or language (see S1 File for complete strategy). Searches comprised of two concepts: socioeconomic consequences and orthopaedic trauma. Keywords were used in combination with database-specific terminology. The reference lists of the included studies were examined for additional papers.

### Screening and assessment of eligibility and data extraction

DistillerSR (Evidence Partners, Ottawa, ON), an online reference management system for systematic reviews, was utilized for screening and study selection. All screening forms were pre-designed and piloted. Two reviewers independently reviewed the titles and abstracts of articles identified in the literature search. All conflicts were included in the full-text screening. The remaining full-text articles were reviewed in a similar independent and duplicate fashion with

two reviewers to determine final inclusion. Any disagreements were resolved through a consensus meeting. When English versions of the articles were unavailable, *Google Translate* (Mountain View, CA) was used to translate the article text into English. Articles that met the full inclusion criteria were used for data extraction. Study characteristics and the demographics, injury characteristics, and socioeconomic outcomes of the study participants were recorded for each included study. As the duration from injury to the socioeconomic assessment was often provided for multiple time points, the outcome and time point were extracted in tandem.

### Quality assessment

The quality of the included studies was assessed following four criteria from the *Users' Guides to the Medical Literature to* evaluate the risk of bias [10]. The criteria included, 1) the duration of follow-up, 2) the proportion of enrolled patients that completed full follow up, 3) a well-described and consistently applied assessment of the socioeconomic outcome, and 4) a study sample with broad eligibility criteria to be considered representative of the fracture population of study. Two reviewers independently assessed the risk of bias. Articles were considered to have a low risk of bias if the study included a representative population, a well-defined socioeconomic outcome, and more than 80% follow-up at least 12-months from injury. Studies were categorized as a high-risk of bias with non-representative samples, ill-defined socioeconomic outcomes, and follow-up rates of less than 70%.

### Data synthesis and analysis

The characteristics of the included studies, the study participants, and the socioeconomic outcomes were described using counts and proportions. The types of fractures were defined using the Arbeitsgemeinschaft für Osteosynthesefragen (AO)/ Orthopaedic Trauma Association (OTA) Fracture and Dislocation Classification Compendium, 2018 [11]. When possible, socioeconomic outcomes were pooled using the inverse variance method and summarize with point estimates with 95% confidence intervals. Given the tremendous heterogeneity in the pooled data ($I^2 > 80\%$), random-effects meta-analyses were performed. Multiple imputations were used to calculate the variance for absenteeism days from work in studies with no measure of variance reported. Cost data were converted from the reported currency to US dollars (USD) based on the market exchange rate on January 1 in the year of publication.

## Results

### Study characteristics

A total of 3,404 titles and abstracts, and subsequently, 972 full-text articles were screened; 205 met our eligibility criteria and were included in the review (Fig 1). The included studies predominantly comprised of retrospective cohort studies (35.6%) and case series (31.7%) (Table 1). The majority of the studies were performed at a single site (78.0%) with a median sample size of 62 patients (IQR: 34–145), and over half were conducted in either Europe (37.6%) or North America (27.3%). In the included prospective studies, the median follow-up was 12 months (IQR: 6–24 months). Retrospective studies had a median follow-up of 18 months (IQR: 12–25). Fractures of the tibia (31.2%) and hand (31.2%) were the most commonly studied. While calcaneus (n = 30), scaphoid (n = 24), and malleolus (n = 18) were the most frequently included fracture locations in the included studies. Over 80% of the included studies were published from 2000 through 2019.

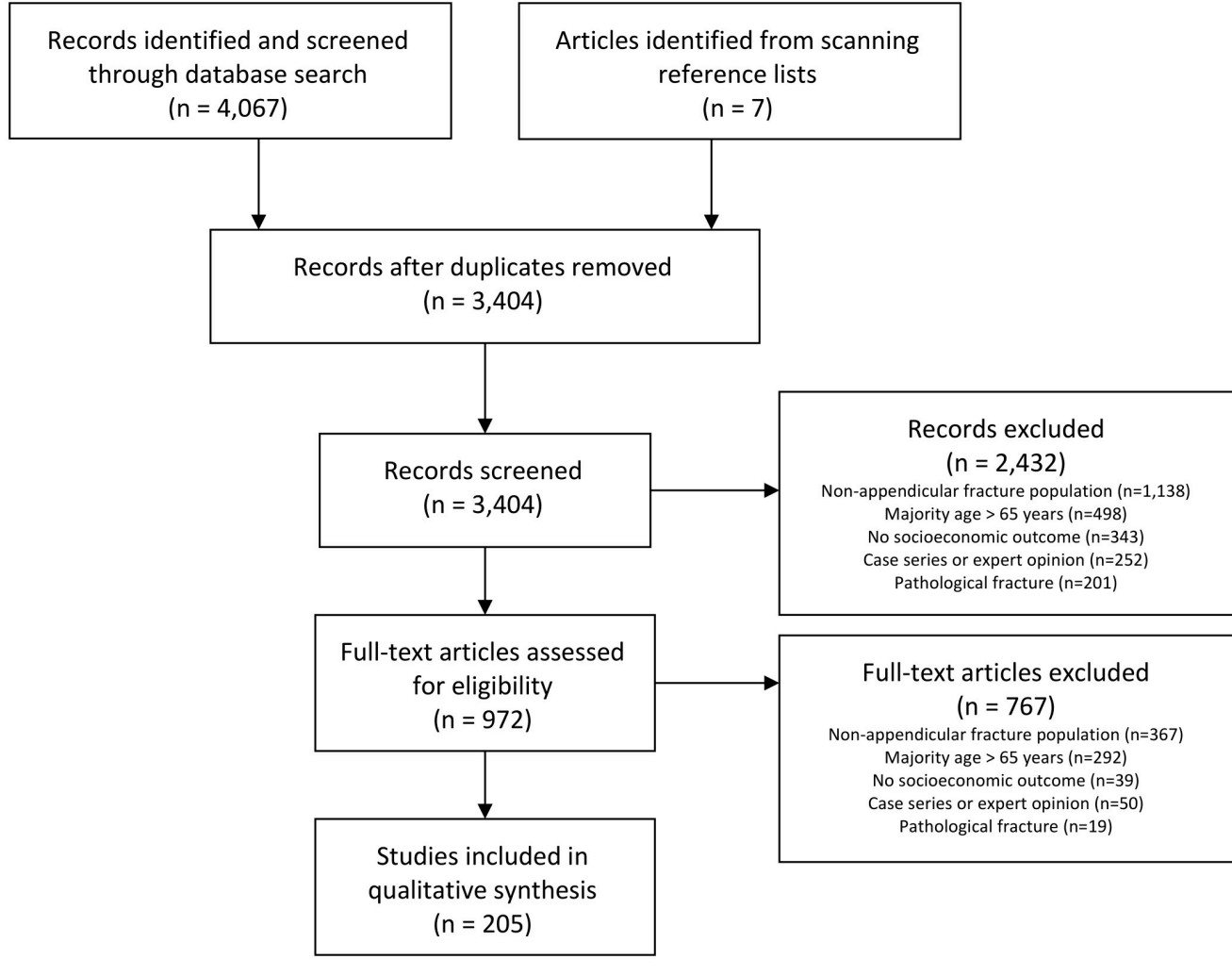

**Fig 1. PRISMA flow chart.**

## Participant characteristics

The 205 studies included 273,618 patients. The mean age of the study participants was 39.8 years (95% CI: 38.1–41.5), and 73.3% were male (95% CI: 71.0–75.4) (Table 2). In the studies that reported the mechanism of injury (n = 115), 75.0% (95% CI: 71.3–78.3) of the study participants had high-energy injuries. The majority of the patients in the included studies were employed at the time of injury (95.0%, 95% CI: 93.9–95.9).

## Socioeconomic outcome measure

Five common socioeconomic outcomes were identified in the included studies (Table 3). The most common outcome measure was return to work (n = 119), closely followed by absenteeism days from work (n = 104). Productivity loss (n = 11), income loss (n = 11), and unemployed due to injury (n = 10) appeared less frequently.

## Return to work

Based on the included literature, return to work measures the proportion of study participants that return to employment at a defined time interval or within the duration of the study.

**Table 1. Summary of study characteristics (n = 205).**

| Study Characteristic | | | No. (%) |
|---|---|---|---|
| Study type | | | |
| | Randomized Controlled Trial | | 25 (12.2) |
| | Prospective Cohort | | 32 (15.6) |
| | Retrospective Cohort | | 73 (35.6) |
| | Case-Control | | 2 (1.0) |
| | Case Series | | 65 (31.7) |
| | Other[a] | | 8 (3.9) |
| Fracture location of study[b] | | | |
| | Humerus | | 41 (20.0) |
| | Forearm | | 36 (17.6) |
| | Femur | | 31 (15.2) |
| | Tibia | | 64 (31.2) |
| | Pelvis | | 24 (11.7) |
| | Hand | | 64 (31.2) |
| | Foot | | 59 (28.8) |
| Continent[c] | | | |
| | Europe | | 77 (37.6) |
| | North America | | 56 (27.3) |
| | Asia | | 39 (19.0) |
| | Australia/New Zealand | | 22 (10.7) |
| | Africa | | 6 (2.9) |
| | South America | | 4 (2.0) |
| | Multi-continent | | 1 (0.05) |
| Number of study sites | | | |
| | Single Site | | 160 (78.0) |
| | Multisite | | 32 (15.6) |
| | Payer Database | | 13 (6.3) |
| Study sample size | | | |
| | 11–50 | | 80 (39.0) |
| | 51–100 | | 59 (28.8) |
| | 101–250 | | 34 (16.6) |
| | 251–500 | | 12 (5.9) |
| | > 500 | | 20 (9.8) |
| Duration of enrollment | | | |
| | Prospective Studies | | |
| | | < 1 year | 9 (15.7) |
| | | 1–3 years | 19 (33.3) |
| | | 4–5 years | 9 (15.7) |
| | | > 5 years | 4 (7.0) |
| | | Not reported | 16 (28.1) |
| | Retrospective Studies | | |
| | | < 1 year | 12 (8.1) |
| | | 1–3 years | 19 (12.8) |
| | | 4–5 years | 39 (26.4) |
| | | > 5 years | 57 (38.5) |
| | | Not reported | 21 (14.2) |
| Length of follow-up, months, median (range) | | | |

*(Continued)*

**Table 1.** (Continued)

| Study Characteristic | | | No. (%) |
|---|---|---|---|
| | Prospective Studies | | |
| | | 0–6 months | 17 (29.8) |
| | | 7–12 months | 23 (40.4) |
| | | 13–24 months | 12 (21.1) |
| | | 25–60 months | 0 (0) |
| | | > 60 months | 4 (7.0) |
| | | Not reported | 1 (1.8) |
| | Retrospective Studies | | |
| | | 0–6 months | 16 (10.8) |
| | | 7–12 months | 40 (27.0) |
| | | 13–24 months | 42 (28.4) |
| | | 25–60 months | 27 (18.2) |
| | | > 60 months | 6 (4.1) |
| | | Not reported | 17 (11.5) |
| Year of publication | | | |
| | 1960–1969 | | 3 (1.5) |
| | 1970–1979 | | 2 (1.0) |
| | 1980–1989 | | 7 (3.4) |
| | 1990–1999 | | 28 (13.7) |
| | 2000–2010 | | 73 (35.6) |
| | 2010–2017 | | 92 (44.9) |

[a] Other study types included four quasi-experimental studies, two longitudinal studies, and two cost-effectiveness studies.

[b] Cumulative total is greater than 100% as 37 studies included more than one fracture location.

[c] Continent refers to where the study was conducted; if not reported explicitly, the location of the corresponding author's institution was used as a proxy.

Several studies broadened the definition to include return to work or participation in an education program. Studies of military populations typically refer to return to duty. Return to work within six months of injury (24.5%) or 12 months of injury (26.1%) were the most common time intervals utilized by the included studies. However, nearly half of the studies did not define a specific time interval for measuring the return to work. Few studies specified if there were any changes in the employer or the work duties for the study participant upon returning to work. These data were mostly obtained using primary data collection (79.8%). Pooled estimates for return to work remained relatively consistent across the 6-, 12-, and 24-month reporting point estimates of 58.7%, 67.7%, and 60.9%, respectively. Thirty-two studies used return to work as the primary outcome.

## Absenteeism days from work

Absenteeism days from work was the second most common socioeconomic outcome in the reviewed studies (n = 104). This outcome was synonymously reported as days lost, time to return to work, temporary disability days, and sick leave. Eleven studies used absenteeism days from work as the primary outcome, and data were predominantly obtained through primary data collection (86.5%). The pooled estimate for mean days absent was 102.3 days (95% CI: 94.8–109.8). Six fracture locations (distal radius, scaphoid, metacarpal, phalanges, malleolus, and calcaneus) had more than five studies that used absenteeism days from work as an outcome, enabling a comparison in the heterogeneity of days absent from employment across

**Table 2. Summary of patient characteristics from included studies (n = 273,618).**

| Characteristic | | No. (%) |
|---|---|---|
| % Male | | |
| | 0–49.9 | 16 (7.8) |
| | 50–74.9 | 74 (36.1) |
| | 75–100 | 90 (43.9) |
| | Not reported | 21 (10.2) |
| Age, mean, years | | |
| | 18–29 | 23 (11.2) |
| | 30–39 | 83 (40.5) |
| | 40–49 | 61 (29.8) |
| | 50–65 | 8 (3.9) |
| | Not reported | 27 (13.2) |
| % Mechanism of injury | | |
| | > 50% high energy | 92 (44.9) |
| | > 50% low energy | 22 (11.2) |
| | Not reported | 90 (43.9) |
| % Employed at baseline | | |
| | 0–49 | 6 (2.9) |
| | 50–74 | 23 (11.2) |
| | 75–89 | 30 (14.6) |
| | 90–100 | 123 (60.0) |
| | Not reported | 23 (11.2) |

those fracture locations. As highlighted in Fig 2, we observed substantially more absenteeism days for study participants with calcaneus fractures than what was observed for study participants with other fracture locations.

## Productivity loss

Of the five main socioeconomic measure, the calculation and reporting of productivity loss had the greatest variation. Several studies used techniques to estimate a monetary value for lost productivity. MacKenzie et al. used the Work Limitations Questionnaire [73], and another study applied an actuarial assessment of impairment due to injury to their study population [79]. Other studies qualitatively assessed lost productivity. Of the 11 studies that assessed productivity loss, three used the metric as their primary outcome. Only one study defined a time interval for their assessment and over a third of the studies collected these data from an existing database.

## Income loss

Income loss was used as a socioeconomic outcome in 11 of the included studies. The outcome was commonly calculated as days absent multiplied by average wage rates in the jurisdiction or the wage cost using public insurance databases [47, 135]. The majority (72.7%) did not specify a time interval for this outcome. The mean lost income for 6-, 12-, and 24-months post-injury was $96, $1,823, and $14,621, respectively. For studies with undefined time intervals, the pooled mean income loss was $3,611 (95% CI: 1,617–5,606). One of the included studies used income loss as their primary outcome.

**Table 3. Summary of socioeconomic outcome measures from the included studies.** The outcomes are described by follow-up time frames commonly associated with various socioeconomic measures, and the practices employed for collecting socioeconomic metrics.

| Outcome | | Return to work (duty) | Absenteeism days from work | Productivity loss | Income loss (USD) | Injury-related unemployment |
|---|---|---|---|---|---|---|
| No. of studies | | 119 [12–130] | 104 [19, 20, 26, 28, 37, 38, 40, 44, 46, 47, 55, 60, 62, 66, 73, 74, 77, 79, 83, 94, 100, 103, 106, 110, 112, 114, 119, 118, 131–206] | 11 [51, 60, 73, 79, 89, 116, 134, 141, 207–209] | 11 [19, 37, 47, 51, 89, 135, 143, 163, 186, 210, 211] | 10 [16, 60, 62, 72, 73, 77, 107, 186, 211, 212] |
| No. of patients | | | | | | |
| | 11–50 | 49 (41.1) | 46 (44.2) | 1 (9.1) | 3 (27.3) | 3 (30.0) |
| | 51–100 | 34 (28.6) | 29 (27.9) | 3 (27.3) | 3 (27.3) | 3 (30.0) |
| | 101–250 | 15 (12.6) | 20 (19.2) | 2 (18.2) | 4 (36.4) | 1 (10.0) |
| | 251–500 | 11 (9.2) | 2 (1.9) | 1 (9.1) | 0 (0) | 2 (20.0) |
| | > 500 | 10 (8.4) | 7 (6.7) | 4 (36.4) | 1 (9.1) | 1 (10.0) |
| No. of studies where the socioeconomic measure was the primary outcome | | 32 (26.9) | 11(10.6) | 3 (27.3) | 1 (9.1) | 0 (0) |
| No. of studies that included each time point* | | | | | | |
| | 0–6 months | 29 (24.5) | - | 1 (9.1) | 1 (9.1) | 1 (10.0) |
| | 7–12 months | 31 (26.1) | - | - | 1 (9.1) | 2 (20.0) |
| | 13–24 months | 20 (16.8) | - | - | 1 (9.1) | 1 (10.0) |
| | > 24 months | 3 (2.5) | - | - | | 1 (10.0) |
| | Undefined | 54 (45.4) | - | 10 (90.9) | 8 (72.7) | 5 (50.0) |
| Point estimate for each time point | | | | | | |
| | 6 months | 58.8% (48.8–68.1)[a] | - | No consistent measure used for productivity loss | $96.0 (-) | 46.2% |
| | 12 months | 67.7% (61.0–73.7)[b] | - | | $1,823.0 (-) | 40.5% (8.4–83.4)[e] |
| | 24 months | 60.9% (51.8–69.3)[c] | - | | $14,621.0 (-) | 42.2% |
| | Undefined | | 102.3 days (94.8–109.8)[d] | | $3,611 (1,617–5,605) | 13.1% (4.8–30.7)[f] |
| Data collection methods | | | | | | |
| | Primary | 95 (79.8) | 90 (86.5) | 4 (36.4) | 4 (36.4) | 8 (80.0) |
| | Database | 18 (15.1) | 13 (12.5) | 4 (36.4) | 7 (63.6) | 2 (20.0) |
| | Not specified | 6 (5.0) | 1 (1.0) | 3 (27.3) | 0 (0) | 0 (0) |
| Risk of bias | | | | | | |
| | High | 12 (10.1) | 8 (7.7) | 1 (9.1) | 0 (0) | 1 (10.0) |
| | Moderate | 96 (80.7) | 87 (83.7) | 9 (81.8) | 9 (81.8) | 7 (70.0) |
| | Low | 12 (9.2) | 9 (8.7) | 1 (9.1) | 2 (18.2) | 2 (20.0) |

[a] $I^2$ = 97.0% (95% CI: 96.2–97.6)

[b] $I^2$ = 95.1% (95% CI: 93.9–96.1)

[c] $I^2$ = 97.5% (95% CI: 96.8–98.0)

[d] $I^2$ = 99.9% (95% CI: 99.9–99.9)

[e] $I^2$ = 97.9% (95% CI: 94.9–99.1)

[f] $I^2$ = 89.1% (95% CI: 77.2–94.8)

* Many studies collected and reported outcome data at multiple time points.

USD = US dollars. Non-US currencies were converted to US dollars based on the exchange rate on January 1 in the publication year. Costs remain nominal for the publication year and were not adjusted for inflation.

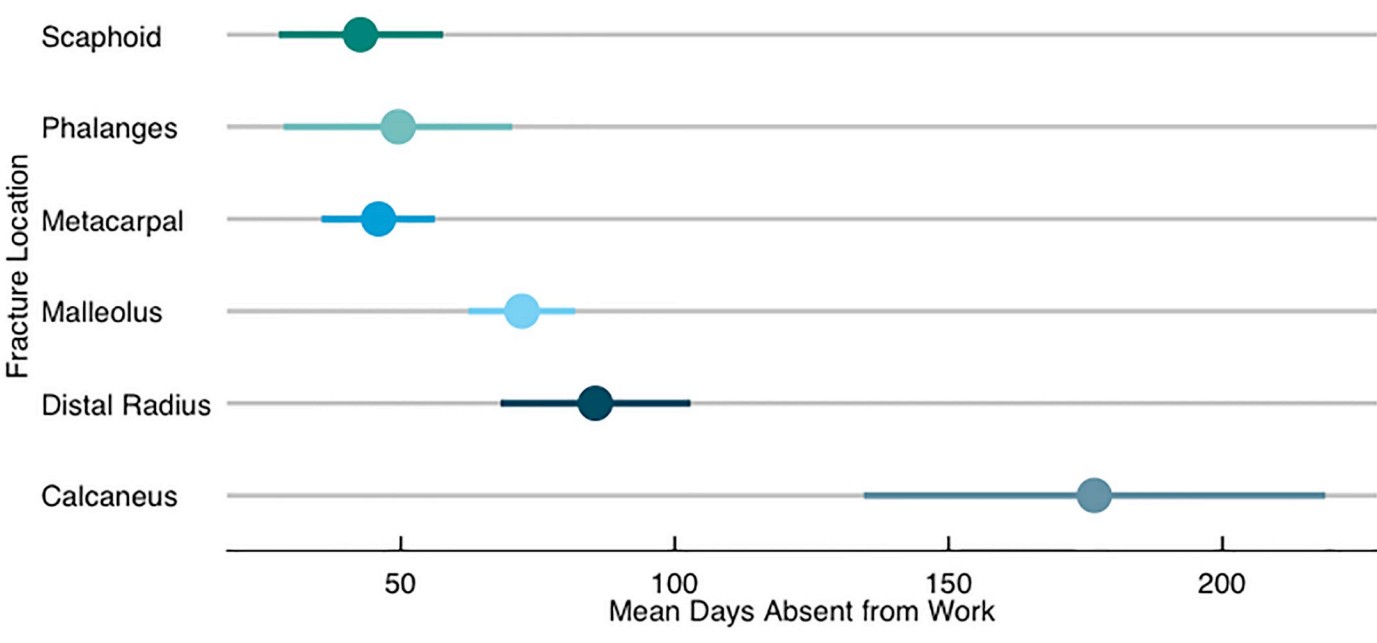

**Fig 2. Mean absenteeism days by fracture location.**

### Injury-related unemployment

Ten of the included studies used injury-related unemployment, or lost employment, as a study outcome. Injury-related unemployment was often described as a level of disability resulting in a withdrawal from the workforce. This measure was predominately determined through primary data collection, and half of the studies did not specify a time interval for the outcome. The pooled proportion of patients that were employed prior to injury but no longer employed at 12-months post-injury was 40.5% (95% CI: 8.4–83.4). For included studies with an undefined time interval, the pooled proportion of lost employment following injury was 13.1% (95% CI: 4.8–30.7).

### Other socioeconomic outcomes

Several other socioeconomic outcome measures were described in the included literature, such as the Sickness Impact Profile, or the Olerud and Molander Score [78, 116]. The accumulation of debt and accessing social assistance were also reported in the literature [118, 211]. Ioannou et al. measured financial worry relative to physical and mental recovery after injury [129]. Finally, Hou et al. integrated health-related quality of life with sick leave days to create a novel measure of health-adjusted leave days [160].

### Risk of bias

Based on our defined criteria, the methodological safeguards against the risk of bias were limited among the included studies. Eighteen of the included studies (8.9%) were categorized as a high risk of bias, while 171 studies were considered to be at moderate risk of bias (83.4%) (Table 4). The main factors leading to an elevated risk of bias were due to inconsistent or lacking definitions of the socioeconomic outcome (71.2%), narrow eligibility criteria (41.0%), and six months or less of follow-up (12.2%). Sixteen of the included studies (7.8%) were deemed to be at low risk of bias.

**Table 4. Risk of bias assessment for the included studies.**

| Assessment Criteria | | | Bias Risk | No. (%) |
|---|---|---|---|---|
| Duration of follow up | | | | |
| | 0–6 months | | High | 33 (16.1) |
| | 7–12 months | | Moderate | 48 (23.4) |
| | 13–24 months | | Low | 48 (23.4) |
| | > 24 months | | Low | 85 (41.5) |
| Proportion of sample that completed full follow-up | | | | |
| | > 90% follow up | | Low | 116 (56.6) |
| | 80–90% follow up | | Low | 28 (13.7) |
| | 70–80% follow up | | Moderate | 11 (5.4) |
| | < 70% follow up | | High | 33 (16.1) |
| | Not reported | | High | 17 (8.3) |
| Described and consistently applied definition of socioeconomic outcome | | | | |
| | Well-described, consistently applied | | Low | 59 (28.7) |
| | Inconsistent or lacking description | | High | 146 (71.2) |
| Sample representative of studied fracture population | | | | |
| | Broad eligibility criteria | | Low | 121 (59.0) |
| | Narrow eligibility criteria | | High | 84 (41.0) |

## Discussion

Orthopaedic trauma can have a profound socioeconomic impact on patients, particularly within a year of injury. Based on the included studies, one-third of patients had not returned to work at one-year post-injury and, on average, patients missed over 100 days of work following their fracture. Data on the long-term socioeconomic impact of orthopaedic trauma is limited but suggests that 13% of fracture patients may lose employment due to injury.

Various measures have been used to quantify the economic impact of orthopaedic trauma. Return to work and absenteeism days from work were the most commonly used socioeconomic outcomes. Productivity loss, income loss, and lost employment were used with much less frequency. Primary data collection was used to capture the socioeconomic outcomes in over three-quarters of the included studies. The majority of the included prospective studies calculated their socioeconomic measures at one year or less from injury. However, even in retrospective studies, over one-third measured their socioeconomic outcomes within one-year of injury. The bias assessment concluded that the methods for measuring the socioeconomic outcomes were vague or lacking entirely in three-quarters of the included studies. Tremendous heterogeneity was observed in the pooled socioeconomic outcomes.

The increased availability of large registry data presents an opportunity for long-term, population-level estimates of the socioeconomic effects of fractures. However, to realize this opportunity, socioeconomic data must be routinely and reliably collected in health data registries, or health registry data must include identifiers that can be linked to available socioeconomic data.

The results of this review identified opportunities to improve the societal relevance of orthopaedic trauma research by demonstrating the limitations in the current approaches of commonly used socioeconomic outcomes. Socioeconomic recovery following injury can be very nuanced, and applying only a single measure of socioeconomic recovery yields inherent bias. Absenteeism days from work fails to describe study participants that do not return to work or return with impairment. Return to work rarely accounts for changes in the employment situation or productivity of the study participants [36]. Productivity loss is difficult to compare across study participants and can be confounded by baseline productivity. Income

loss is largely dependent on the pre-injury income distribution of the study population. As study duration increases, new unemployment tends to be a rare outcome for most types of fractures and is easily confounded by the type of pre-injury employment.

Many of the included studies highlight practical approaches to measuring socioeconomic impact. Several of the included studies, such as those by MacKenzie et al. and Gardner et al. [73, 155], utilized a multifaceted approach to assessing the socioeconomic outcomes for the study population. Mortelmans et al. combine absenteeism days from work and an estimate of impairment for a detailed understanding of the socioeconomic outcomes following an intraarticular calcaneus fracture [79]. However, the specific method for quantifying impairment lacks description. Nusser et al. added a minimum duration of work absence to their socioeconomic outcome reporting [86]. Several other studies specifically characterized the sustained absence from work into categories such as retired, unemployed, undergoing rehabilitation, recipient of disability payments, in school, never working, or retraining for a different job [85, 115]. Prognostic modeling and stratified analysis included in five studies highlight several common confounders, such as the physical demands of the pre-injury employment [77, 79, 139, 148, 18, 95]. Additionally, the association between study participant age and return to work as well as the association between having dependents and return to work were identified and should be investigated as confounders in future studies on the socioeconomic consequences of extremity fractures [66, 93].

The systematic review and meta-analysis included a broad range of extremity fracture research from 40 countries and strictly adhered to the PRISMA guideline for conduct and reporting. However, despite these strengths, there were several limitations. Socioeconomic outcomes were reported at inconsistent time intervals in the included studies, therefore limiting our ability for both pooled and subgroup analyses. Other subgroup analyses were not possible due to inconsistent reporting of potential confounders, such as the severity of the injury, patient comorbidities, the type of pre-injury employment, and legal adjudication for compensation. All of these factors are likely to affect the patient's post-injury economic well-being. The assessment of study generalizability and a consistent socioeconomic outcome definition used in our risk of bias assessment carries a level of subjectivity. However, the appraisal was performed in duplicate. Finally, the described socioeconomic measure does not represent a fully inclusive list; rather, it includes those socioeconomic outcomes currently being utilized in orthopaedic trauma research. There are likely other socioeconomic outcomes, such as the Work Productivity and Activity Impairment questionnaire [213], that are available but were not utilized by the included studies.

Determining the effect of orthopaedic trauma on the economic well-being of the patient is essential for designing value-based care programs. In addition, these data inform surgeon-patient communication on recovery expectations, support the prioritization of health policies, and inform the design of future therapeutic studies aimed at mitigating the socioeconomic consequences of injury. The findings of this meta-analysis suggest that orthopaedic trauma can have a substantial socioeconomic impact on patients, and therefore also affect a person's psychological well-being and happiness. However, the current techniques to measure socioeconomic outcomes following orthopaedic trauma are widely varied in both design and implementation. Informative and accurate socioeconomic outcome assessment requires a multifaceted approach and further standardization.

## Supporting information

**S1 File. Detailed search strategies.**
(DOCX)

**S1 Checklist. PRISMA checklist.**
(DOC)

**S1 Dataset. Dataset used for meta-analysis.**
(CSV)

## Acknowledgments

We thank Ms. Emilie Ludeman, MSLIS, at the University of Maryland School of Medicine, for her assistance in developing the search strategy and performing the search.

## Author Contributions

**Conceptualization:** Nathan N. O'Hara, Gerard P. Slobogean, Niek S. Klazinga.

**Data curation:** Nathan N. O'Hara, Marckenley Isaac.

**Formal analysis:** Nathan N. O'Hara, Marckenley Isaac.

**Investigation:** Nathan N. O'Hara, Marckenley Isaac.

**Methodology:** Nathan N. O'Hara, Marckenley Isaac, Gerard P. Slobogean.

**Project administration:** Nathan N. O'Hara, Marckenley Isaac, Gerard P. Slobogean, Niek S. Klazinga.

**Resources:** Nathan N. O'Hara, Gerard P. Slobogean, Niek S. Klazinga.

**Software:** Nathan N. O'Hara.

**Supervision:** Gerard P. Slobogean, Niek S. Klazinga.

**Writing – original draft:** Nathan N. O'Hara.

**Writing – review & editing:** Nathan N. O'Hara, Marckenley Isaac, Gerard P. Slobogean, Niek S. Klazinga.

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
