## [Decision Letter · Decision Letter 0]

26 Nov 2019

PONE-D-19-30486

The Socioeconomic Impact of Orthopaedic Trauma: A Systematic Review and Meta-Analysis

PLOS ONE

Dear Mr. O'Hara,

Thank you for submitting your manuscript to PLOS ONE. After careful consideration, we feel that it has merit but does not fully meet PLOS ONE’s publication criteria as it currently stands. Therefore, we invite you to submit a revised version of the manuscript that addresses the points raised during the review process.

We would appreciate receiving your revised manuscript by Jan 10 2020 11:59PM. To enhance the reproducibility of your results, we recommend that if applicable you deposit your laboratory protocols in protocols.io, where a protocol can be assigned its own identifier (DOI) such that it can be cited independently in the future. For instructions see: http://journals.plos.org/plosone/s/submission-guidelines#loc-laboratory-protocolsPlease include the following items when submitting your revised manuscript:A rebuttal letter that responds to each point raised by the academic editor and reviewer(s). This letter should be uploaded as separate file and labeled 'Response to Reviewers'.A marked-up copy of your manuscript that highlights changes made to the original version. This file should be uploaded as separate file and labeled 'Revised Manuscript with Track Changes'.An unmarked version of your revised paper without tracked changes. This file should be uploaded as separate file and labeled 'Manuscript'.

We look forward to receiving your revised manuscript.

Kind regards,

Osama Farouk

Academic Editor

PLOS ONE

Journal Requirements:

2. Please ensure you have performed the latest search within the past 12 months, the current search is out of date.

Additional Editor Comments (if provided):

Reviewers' comments:

Reviewer's Responses to Questions

**Comments to the Author**

1. Is the manuscript technically sound, and do the data support the conclusions?

Reviewer #1: Yes

Reviewer #2: Yes

2. Has the statistical analysis been performed appropriately and rigorously? 

Reviewer #1: Yes

Reviewer #2: Yes

3. Have the authors made all data underlying the findings in their manuscript fully available?

Reviewer #1: Yes

Reviewer #2: Yes

4. Is the manuscript presented in an intelligible fashion and written in standard English?

Reviewer #1: Yes

Reviewer #2: Yes

5. Review Comments to the Author

Reviewer #1: General comments:

The study idea about the socioeconomic impact of orthopaedic trauma is interesting and valuable, furthermore to be studied in a systematic review and meta- analysis study design elaborates more information in a wider and fruitful way.

In general, it’s a well written manuscript according to the PRISMA guidelines. I have provided some remarks below.

Abstract:

- The abstract is informative but the number of the included studies is not mentioned, please add it.

Methods:

Under: Data Synthesis and Analysis section, in line 122: linguistic correction; “ fractures types” to be corrected to fractures’ types or types of fractures.

Results:

All tables and figures are presented in a clear and informative way, with few comments:

- Table (1): in study types: please clarify in details the other types and insert it in the table or as a footnote under the table.

- One of the study characteristics included is fracture location of the study, in some studies more than one location were found. This characteristic is better to be included under patient characteristics as patient wise, to be included in table (2) after mechanism of injury.

- In table (3) about Socioeconomic Outcome Measures, under data collection methods: clarify in details the details in the table or as a footnote.

- The term “days absent from work” could be replaced by absenteeism days from work all over the manuscript.

Discussion:

- Is well written and covering all items.

- In line 309: What’s meant by prohibited? Please, replace with suitable term.

Reviewer #2: The Manuscript The Socioeconomic Impact of Orthopaedic Trauma: A Systematic Review and Meta-

Analysis addresses an extremely important issue of orthopedic health care: although due to aging societies, age related diseases are more and more frequent, orthopedic Trauma is jeopardizing patients in their best years. By that the quantitative description of work loss etc. is for outstanding interest for the scientific Society and I strongly recommend to publish this study. My only Question would be an Annotation in the discussion: the follow up periods are significantly different between prospective and retrospective studies, which somehow reflects a currently taking place paradigma Change in the operative disciplines a Little bit away from the prospective studies towards the large retrospective Register studies. Hence, it would be good to add one or two sentetnce in the discussion, that the Long term follow ups in orthopedic surgery require probably more big Register studies in the future. Beside this, tha manuscript has my full support for publication and I want to congratulate the authors to their work

6. PLOS authors have the option to publish the peer review history of their article (what does this mean?). If published, this will include your full peer review and any attached files.

Reviewer #1: Yes: Dalia G Mahran

Professor of Public Health and Community Medicine, Faculty of Medicine, Assiut University, Assiut, Egypt

Reviewer #2: Yes: Peter Biberthaler, MD

---

## [Author Response · Author response to Decision Letter 0]

20 Dec 2019

Response to Reviewers

Journal Requirements:

Response: We have adjusted the file naming to comply with PLOS ONE’s requirements.

2. Please ensure you have performed the latest search within the past 12 months, the current search is out of date.

Response: We have updated the search on December 3, 2019. The revised search has added 15 new articles to the systematic review and meta-analysis but did not qualitatively change the findings of the study.

Revision: An experienced academic research librarian conducted searches in MEDLINE (Ovid), Embase (Elsevier), and Scopus on December 3, 2019, without restrictions on publication date or language (see S1 for complete strategy). [Methods]

A total of 3,404 titles and abstracts, and subsequently, 972 full-text articles were screened; 205 met our eligibility criteria and were included in the review (Fig 1). [Results]

Reviewers' comments:

Reviewer #1: General comments:

The study idea about the socioeconomic impact of orthopaedic trauma is interesting and valuable, furthermore to be studied in a systematic review and meta- analysis study design elaborates more information in a wider and fruitful way.

In general, it’s a well written manuscript according to the PRISMA guidelines. I have provided some remarks below.

Response: We thank the reviewer for their complimentary remarks.

Abstract:

- The abstract is informative but the number of the included studies is not mentioned, please add it.

Response: Thank you for identifying this oversight. We have added the number of included studies in the revised manuscript.

Revision: Two-hundred-five studies met the eligibility criteria. [Abstract]

Methods:

Under: Data Synthesis and Analysis section, in line 122: linguistic correction; “ fractures types” to be corrected to fractures’ types or types of fractures.

Response: We have corrected the sentence as suggested.

Revision: The types of fractures were defined using the Arbeitsgemeinschaft für Osteosynthesefragen (AO)/ Orthopaedic Trauma Association (OTA) Fracture and Dislocation Classification Compendium, 2018 [11].

Results:

All tables and figures are presented in a clear and informative way, with few comments:

- Table (1): in study types: please clarify in details the other types and insert it in the table or as a footnote under the table.

Response: We have added the study types included in the “other” category as a footnote to Table 1.

Revision: Other study types included four quasi-experimental studies, two longitudinal studies, and two cost-effectiveness studies.

- One of the study characteristics included is fracture location of the study, in some studies more than one location were found. This characteristic is better to be included under patient characteristics as patient wise, to be included in table (2) after mechanism of injury.

Response: We agree that the fracture location would be more informative as patient-level data. However, many of the studies had broad eligibility criteria that included several fracture locations but did not report the number of patients that had fractures in each specific location in their manuscript. We were, therefore, unable to devise precise estimates on the number of individual patients that sustained specific fractures in our meta-analysis dataset.

- In table (3) about Socioeconomic Outcome Measures, under data collection methods: clarify in details the details in the table or as a footnote.

Response: The “other” type of data collection method should have been more accurately coded as “not specified”. We have updated the table accordingly.

- The term “days absent from work” could be replaced by absenteeism days from work all over the manuscript.

Response: We thank the reviewer for their suggestion and have updated “days absent from work” to “absenteeism days from work” throughout the manuscript.

Discussion:

- Is well written and covering all items.

- In line 309: What’s meant by prohibited? Please, replace with suitable term.

Response: We have changed “prohibited” to “not possible” in the discussion section.

Revision: Other subgroup analyses were not possible due to inconsistent reporting of potential confounders, such as the severity of the injury, patient comorbidities, the type of pre-injury employment, and legal adjudication for compensation.[Discussion]

Reviewer #2: The Manuscript The Socioeconomic Impact of Orthopaedic Trauma: A Systematic Review and Meta-Analysis addresses an extremely important issue of orthopedic health care: although due to aging societies, age related diseases are more and more frequent, orthopedic Trauma is jeopardizing patients in their best years. By that the quantitative description of work loss etc. is for outstanding interest for the scientific Society and I strongly recommend to publish this study. 

Response: We thank the reviewer for their gracious feedback.

My only Question would be an Annotation in the discussion: the follow up periods are significantly different between prospective and retrospective studies, which somehow reflects a currently taking place paradigma Change in the operative disciplines a Little bit away from the prospective studies towards the large retrospective Register studies. Hence, it would be good to add one or two sentetnce in the discussion, that the Long term follow ups in orthopedic surgery require probably more big Register studies in the future. Beside this, tha manuscript has my full support for publication and I want to congratulate the authors to their work

Response: We agree with the reviewer that registries present an incredible opportunity to obtain long-term, population-level estimates of the socioeconomic effects of fractures. We have added that important point to the discussion, but also included the caveat that to estimate socioeconomic effects, health registry data must reliably collect socioeconomic measures or have identifiers that can be linked to socioeconomic registries.

Revision: The increased availability of large registry data presents an opportunity for long-term, population-level estimates of the socioeconomic effects of fractures. However, to realize this opportunity, socioeconomic data must be routinely and reliably collected in health data registries, or health data registries must include identifiers than can be linked to socioeconomic registries.

---

## [Decision Letter · Decision Letter 1]

3 Jan 2020

The Socioeconomic Impact of Orthopaedic Trauma: A Systematic Review and Meta-Analysis

PONE-D-19-30486R1

Dear Dr. O'Hara,

We are pleased to inform you that your manuscript has been judged scientifically suitable for publication and will be formally accepted for publication once it complies with all outstanding technical requirements.

With kind regards,

Osama Farouk

Academic Editor

PLOS ONE

Additional Editor Comments (optional):

Reviewers' comments:

Reviewer's Responses to Questions

**Comments to the Author**

1. If the authors have adequately addressed your comments raised in a previous round of review and you feel that this manuscript is now acceptable for publication, you may indicate that here to bypass the “Comments to the Author” section, enter your conflict of interest statement in the “Confidential to Editor” section, and submit your "Accept" recommendation.

Reviewer #1: (No Response)

Reviewer #2: All comments have been addressed

2. Is the manuscript technically sound, and do the data support the conclusions?

Reviewer #1: (No Response)

Reviewer #2: Yes

3. Has the statistical analysis been performed appropriately and rigorously? 

Reviewer #1: (No Response)

Reviewer #2: Yes

4. Have the authors made all data underlying the findings in their manuscript fully available?

Reviewer #1: (No Response)

Reviewer #2: Yes

5. Is the manuscript presented in an intelligible fashion and written in standard English?

Reviewer #1: (No Response)

Reviewer #2: Yes

6. Review Comments to the Author

Reviewer #1: (No Response)

Reviewer #2: The paper

The Socioeconomic Impact of Orthopaedic Trauma: A Systematic Review and Meta-Analysis

is Ready to go, all remarks were adressed, congrats!

7. PLOS authors have the option to publish the peer review history of their article (what does this mean?). If published, this will include your full peer review and any attached files.

Reviewer #1: Yes: Dalia G Mahran

Reviewer #2: Yes: Peter Biberthaler

---

## [Editor Report · Acceptance letter]

8 Jan 2020

PONE-D-19-30486R1 

The Socioeconomic Impact of Orthopaedic Trauma: A Systematic Review and Meta-Analysis 

Dear Dr. O'Hara:

I am pleased to inform you that your manuscript has been deemed suitable for publication in PLOS ONE. Congratulations! Your manuscript is now with our production department. 

With kind regards,

on behalf of

Dr. Osama Farouk 

Academic Editor

PLOS ONE